# Variability in Grading Diabetic Retinopathy Using Retinal Photography and Its Comparison with an Automated Deep Learning Diabetic Retinopathy Screening Software

**DOI:** 10.3390/healthcare11121697

**Published:** 2023-06-09

**Authors:** Chin Sheng Teoh, Kah Hie Wong, Di Xiao, Hung Chew Wong, Paul Zhao, Hwei Wuen Chan, Yew Sen Yuen, Thet Naing, Kanagasingam Yogesan, Victor Teck Chang Koh

**Affiliations:** 1Department of Ophthalmology, National University Health System, Singapore 119228, Singapore; chin_sheng_teoh@nuhs.edu.sg (C.S.T.);; 2Commonwealth Scientific and Industrial Research Organisation, Urrbrae 5064, Australia; 3Medicine Biostatistics Unit, Yong Loo Lin School of Medicine, National University of Singapore, Singapore 119077, Singapore; 4School of Medicine, University of Notre Dame, Fremantle 6160, Australia; 5Centre for Innovation and Precision Eye Health, Yong Loo Lin School of Medicine, National University of Singapore, Singapore 119077, Singapore

**Keywords:** automated screening software, deep learning, diabetes retinopathy, grading, variability

## Abstract

Background: Diabetic retinopathy (DR) screening using colour retinal photographs is cost-effective and time-efficient. In real-world clinical settings, DR severity is frequently graded by individuals of different expertise levels. We aim to determine the agreement in DR severity grading between human graders of varying expertise and an automated deep learning DR screening software (ADLS). Methods: Using the International Clinical DR Disease Severity Scale, two hundred macula-centred fundus photographs were graded by retinal specialists, ophthalmology residents, family medicine physicians, medical students, and the ADLS. Based on referral urgency, referral grading was divided into no referral, non-urgent referral, and urgent referral to an ophthalmologist. Inter-observer and intra-group variations were analysed using Gwet’s agreement coefficient, and the performance of ADLS was evaluated using sensitivity and specificity. Results: The agreement coefficient for inter-observer and intra-group variability ranged from fair to very good, and moderate to good, respectively. The ADLS showed a high area under curve of 0.879, 0.714, and 0.836 for non-referable DR, non-urgent referable DR, and urgent referable DR, respectively, with varying sensitivity and specificity values. Conclusion: Inter-observer and intra-group agreements among human graders vary widely, but ADLS is a reliable and reasonably sensitive tool for mass screening to detect referable DR and urgent referable DR.

## 1. Introduction

Diabetic retinopathy is a leading cause of blindness in the world, affecting a significant portion of the global population [1]. Projections indicate that by 2030, the number of people with diabetes will double from 171 million to 366 million [2]. Among the people with diabetes mellitus, about one-third have diabetic retinopathy, and one-third of these have vision-threatening diabetic retinopathy [2]. The resulting visual loss from diabetic retinopathy imposes a substantial economic burden due to medical costs and productivity losses, particularly among working-age adults [3,4]. The risk of diabetic retinopathy-related vision loss can be reduced by timely detection and treatment [4,5,6,7]. However, limited access to ophthalmic services and the high cost of specialist consultation often act as barriers to early diabetic retinopathy detection [8]. Screening is defined as presumptive identification or the detection of unrecognised disease through the application of rapid tests and examinations [9]. It is important to recognise that screening tests are not intended to provide a definitive diagnosis. Their primary objective is to identify individuals who may be at a high risk of a particular condition, signalling the need for further assessment by a specialist [9]. 

The International Council of Ophthalmology underscores the importance of effective screening for diabetic retinopathy in global diabetes management [10]. However, relying solely on ophthalmologists or retinal subspecialists to screen all diabetic patients is an inefficient use of resources [10]. To address the urgent need for expanded diabetic retinopathy screening coverage and overcome the technical limitations of the current methods and approach, innovative automated deep learning-based technologies have been developed to play a role in the screening of diabetic retinopathy. Automated deep learning-based systems or artificial intelligence-powered screening algorithms have demonstrated remarkable accuracy in identifying individuals who require specialist referral, leading to a substantial decrease in the number of cases necessitating evaluation by an ophthalmologist [11,12].

Macula-centred 45-degree field-of-view colour fundus photography is commonly used for diabetic retinopathy screening. To enhance early detection, diabetic retinopathy screening modalities such as telemedicine [13] and automated deep learning diabetic retinopathy screening software [14,15] have been adopted globally. Telemedicine is an efficient yet labour-intensive screening method for the identification and prevention of diabetic retinopathy. It helps to reduce the burden on congested healthcare facilities while providing critical care access to patients in rural and underserved regions [9]. The rapid growth of automated deep learning diabetic retinopathy screening software is enhancing the diagnosis and treatment of diabetic retinopathy. The employed algorithms have the capacity to analyse vast quantities of retinal images and accurately detect signs of diabetic retinopathy, such as microaneurysms, haemorrhages, and exudates. This technology can assist ophthalmologists in diagnosing and monitoring the progression of diabetic retinopathy, enabling earlier and more effective treatment [14,15]. However, the effectiveness of these diabetic retinopathy screening methods remains debatable [11,12,16,17], with retinal photography still being the preferred screening test for early diabetic retinopathy detection [18]. Typically, the grading of these photographs is performed by non-physicians [19], healthcare workers of different specialties and levels of expertise, including optometrist [16,20], ophthalmologists, endocrinologists [16] and general practitioners [16]. 

Despite the widespread utilisation of retinal photography, there is a paucity of real-world clinical studies evaluating the variability of diabetic retinopathy severity grading among human graders of different levels of expertise and automated deep learning diabetic retinopathy screening software using the same macula-centred fundus photographs. Inter-observer variability can have adverse implications for appropriate referral, potentially leading to treatment delays or unnecessary referrals to ophthalmologists. This study aims to investigate the variability of diabetic retinopathy severity grading among human graders of different specialties and levels of expertise, and the grading agreement with an automated deep learning diabetic retinopathy screening software. Additionally, this study aims to determine the performance of an automated deep learning diabetic retinopathy screening software by examining the sensitivity, specificity, area under receiver operating characteristic curves and accuracy in the detection of non-referable diabetic retinopathy, non-urgent referable diabetic retinopathy, and urgent referral diabetic retinopathy.

## 2. Materials and Methods

We obtained approval from the institution’s ethical board, the Domain Specific Review Board (study reference number: 2017/00488). A total of two hundred diabetic retinopathy macula-centred fundus photographs were randomly selected from patients with diabetes mellitus who attended their diabetic retinopathy screening visits at the outpatient ophthalmology clinic at the National University Hospital of Singapore between January and March 2017. Visucam Pro NM (Carl Zeiss, Germany) was used for capturing the fundus photographs. Grading was performed based on the International Clinical Diabetic Retinopathy Disease Severity Scale [21], classifying the fundus photographs as follows: no diabetic retinopathy, mild/moderate/severe non-proliferative diabetic retinopathy, proliferative diabetic retinopathy, and poor image quality (Figure 1). Poor image quality was attributed to media opacity, such as corneal scars, dense cataracts and small pupils. Non-referable diabetic retinopathy was defined as no diabetic retinopathy or mild non-proliferative diabetic retinopathy without macular abnormality. Non-urgent referable diabetic retinopathy was defined as fundus photographs with moderate non-proliferative diabetic retinopathy or fundus photographs with poor image quality. Fundus photographs were classified as urgent referable diabetic retinopathy if they showed signs that were consistent with those of severe non-proliferative diabetic retinopathy or proliferative diabetic retinopathy. The grading outcomes of the individual grader and the automated deep learning diabetic retinopathy screening software were assessed and compared to those of three retinal specialists, which were used as the reference (ground truth). In the presence of disagreement in diabetic retinopathy severity grading, the reference was set based on the majority of the agreement of the three retinal specialists. However, in the event of no consensus, the reference was set based on the opinion of the most senior retinal specialist.

In this study, the graders consisted of ophthalmologists, family medicine physicians, and medical students (Table 1). The selection of graders was based on their experience with grading diabetic retinopathy severity on colour fundus photographs. The ophthalmologist group comprised three retinal specialists and three second-year ophthalmology residents who were undergoing board-certified ophthalmology residency training. The family medicine physician group included three family medicine physicians with five to eight years of experience in managing diabetic patients. However, this group of family medicine physicians had limited experience in grading diabetic retinopathy using fundus photographs, with less than 10 fundus photographs graded per year. Lastly, the medical student group consisted of two fifth-year students who possessed basic knowledge of diabetic retinopathy but had no prior experience in grading it. No formal training in diabetic retinopathy severity grading was provided to any of the graders before commencing the fundus photograph grading, except for the provision of standard photographs based on the International Clinical Diabetic Retinopathy Disease Severity Scale as a reference. The data collected were then analysed for inter-observer and intra-group variability. Inter-observer variability was defined as the extent of grading agreement variation among the graders compared to the reference. Intra-group variability referred to the variability observed within each group of graders with similar specialties and levels of expertise.

An automated deep learning diabetic retinopathy screening software has been developed by the Commonwealth Scientific and Industrial Research Organisation for grading of diabetic retinopathy. The system includes three algorithms: the fundus photograph quality assessment algorithm, the diabetic retinopathy referral algorithm, and the diabetic retinopathy urgent referral algorithm. The quality assessment algorithm evaluates the quality of a fundus photograph to determine whether or not it is suitable for further diabetic retinopathy grading. If a photograph is considered of good quality, it is then subjected to the diabetic retinopathy referral algorithm and the diabetic retinopathy urgent referral algorithm for further assessment. The diabetic retinopathy referral algorithm classifies fundus photographs into referral and non-referral categories based on the severity of diabetic retinopathy. The referral category includes moderate non-proliferative diabetic retinopathy, severe non-proliferative diabetic retinopathy, and proliferative diabetic retinopathy. In contrast, the non-referral category includes normal fundus photographs or mild non-proliferative diabetic retinopathy. The diabetic retinopathy urgent referral algorithm identifies severe non-proliferative diabetic retinopathy or proliferative diabetic retinopathy for swift medical attention. Convolutional neural networks were used to develop the quality assessment and diabetic retinopathy referral algorithms using the transfer learning approach and inception architecture. The diabetic retinopathy urgent referral algorithm was designed using diabetic retinopathy lesion (microaneurysms, haemorrhages (vitreous, pre-retinal and intra-retina haemorrhages), exudates and new vessels) detection and machine learning-based classification methods. The training and validation were performed using a dataset consisting of 30,000 images and the DiaRetDB database (https://www.kaggle.com/datasets/nguyenhung1903/diaretdb1-v21 (accessed on 16 May 2023)). 

### Statistical Analysis and Its Rationale

The inter-observer and intra-group agreement of the graders were assessed with Gwet’s AC2 agreement coefficient. The results of the agreement coefficient were categorised into different levels of agreement, i.e., (a) <2 represented poor agreement, (b) 0.21 to 0.40 represented fair agreement, (c) 0.41 to 0.60 represented moderate agreement, (d) 0.61 to 0.80 represented good agreement, and (e) >0.80 represented very good agreement [22].

The accuracy, diagnostic ability and performance of the automated deep learning diabetic retinopathy software in identifying different levels of diabetic retinopathy were evaluated using the area under curve of the receiver operating characteristic curve [23]. Specifically, we assessed the automated deep learning diabetic retinopathy software’s ability to identify non-referrable diabetic retinopathy, non-urgent referrable diabetic retinopathy, and urgent referrable diabetic retinopathy. Utilising the area under the receiver operating characteristic curve, we visually represented the software’s performance and discriminative ability across all possible threshold values [23] allowing a straightforward comparison of its performance.

Logistic regression model was used to estimate the predicted probabilities of non-referable diabetic retinopathy, non-urgent referable diabetic retinopathy and urgent referable diabetic retinopathy by risk score. There were three possible categories of the automated deep learning diabetic retinopathy screening software. We assigned a score for the categories of the automated deep learning diabetic retinopathy screening software according to the logistic regression coefficients. 

The receiver operating characteristic curve provides an evaluation of the automated deep learning software’s capability to identify various levels of referrable diabetic retinopathy. In contrast, the Youden index serves as a measure to determine the maximum potential effectiveness [24] of the automated deep learning software at identifying different referable levels of diabetic retinopathy. We calculated the discriminatory ability of the risk score in predicting non-referable diabetic retinopathy, non-urgent referable diabetic retinopathy and urgent referable diabetic retinopathy using the area under the receiver operating characteristic curves with Youden’s index. An area under curve value of >0.70 indicated a fair discriminatory ability in identifying different referrable levels of diabetic retinopathy. The optimal cut-offs were derived from the Youden’s index [24], a statistic test that assess the performance of a dichotomous diagnostic test. It has a value range from −1 to 1, with the value closest to 1 indicating fewer false positives or false negatives. Sensitivity and specificity were calculated based on the optimal cutoffs. Data were analysed using Statistical Analysis Systems (SAS) version 9.4 (SAS Institute Inc., Cary, NC, USA).

## 3. Results

Out of the 200 fundus photographs graded by three retinal specialists, it was found that 31 fundus photographs (15.5%) were graded as having no diabetic retinopathy, while 32 fundus photographs (16.0%) were deemed to have poor image quality. The remaining fundus photographs were graded as mild, moderate and severe non-proliferative diabetic retinopathy with 46 fundus photographs (23.0%), 49 fundus photographs (24.5%) and 32 fundus photographs (16.0%), respectively. There were 10 fundus photographs (5.0%) graded as proliferative diabetic retinopathy.

The degree of inter-observer Gwet’s agreement coefficient values ranged from 0.205 to 0.848, with fair to very good agreement (Table 2). The inter-observer Gwet’s agreement between the retinal specialists, ophthalmology residents, family medicine physicians, undergraduate medical students and automated deep learning diabetic retinopathy screening software against the reference (ground truth) were good to very good, moderate to good, fair to good, fair to moderate and good, respectively. The agreement was significantly (*p*-value < 0.001) higher in the ophthalmology group than that in the non-ophthalmology group. The degree of intra-group Gwet’s agreement coefficient values among retinal specialists, ophthalmology residents, family medicine physicians, and medical students were 0.628 (good agreement), 0.655 (good agreement), 0.536 (moderate agreement) and 0.446 (moderate agreement), respectively.

Area under receiver operating characteristic curves have been utilised as a combined measure of sensitivity and specificity to describe the inherent validity of the automated deep learning diabetic retinopathy screening software [23]. Figure 2, Figure 3 and Figure 4 show the area under receiver operating characteristic curve analysis between the automated deep learning diabetic retinopathy screening software with the reference (ground truth) for non-referable diabetic retinopathy, non-urgent referable diabetic retinopathy and urgent referable diabetic retinopathy. The area under receiver operating characteristic curves of non-referable diabetic retinopathy, non-urgent referable diabetic retinopathy and urgent referable diabetic retinopathy for the automated deep learning diabetic retinopathy screening software were 0.879 (95% confidence interval: 0.832 to 0.926, *p*-value < 0.001), 0.714 (95% confidence interval: 0.648 to 0.781, *p*-value < 0.001) and 0.836 (95% confidence interval: 0.788 to 0.885, *p*-value < 0.001), respectively. 

The risk scores of non-referable diabetic retinopathy were 5, 2 and 0 when the automated deep learning diabetic retinopathy screening software showed non-referable diabetic retinopathy, non-urgent referable diabetic retinopathy and urgent referable diabetic retinopathy, respectively. The risk scores of non-urgent referable diabetic retinopathy were 0, 3 and 2 when the automated deep learning diabetic retinopathy screening software showed non-referable diabetic retinopathy, non-urgent referable diabetic retinopathy and urgent referable diabetic retinopathy, respectively. The risk scores of urgent referable diabetic retinopathy were 0, 25 and 27 when the automated deep learning diabetic retinopathy screening software showed non-referable diabetic retinopathy, non-urgent referable diabetic retinopathy and urgent referable diabetic retinopathy, respectively. 

The optimal cutoff values according to the Youden index for the risk score in predicting non-referable diabetic retinopathy, non-urgent referable and urgent referable diabetic retinopathy were 3 (with a sensitivity of 70.1%, specificity of 93.5%, positive predictive value of 87.1%, and negative predictive value of 83.3%), 2 (with a sensitivity of 90.1%, specificity of 45.4%, positive predictive value of 54.9%, and negative predictive value of 87.1%) and 26 (with a sensitivity of 85.7%, specificity of 75.9%, positive predictive value of 48.6%, and negative predictive value of 95.2%), respectively.

## 4. Discussion

This study analysed locally acquired colour fundus photographs at a tertiary centre in Singapore and revealed a significant range of intra-group and inter-observer variability among human graders with different levels of expertise, particularly in the non-ophthalmology group. Notably, a previous study conducted by Grzybowski A et al. demonstrated high variability in diabetic retinopathy grading among non-trained retina specialists [25]. These findings underscore the need for global improvements in the current grading of diabetic retinopathy severity [25], especially in centres where human graders are predominantly involved. The wide range of inter-observer agreement observed across all groups can be attributed to several factors. Firstly, the graders had varying levels of clinical expertise, experience, and specialties, reflecting the diversity of real-world diabetic retinopathy screening practices. Secondly, the assessment of disease severity was limited by the two-dimensional photographs within a 45-degree field of view compared to that in a dilated fundus examination. Lastly, unlike research settings or dedicated grading centres where graders undergo standardised training and are subjected to active auditing systems to ensure consistent grading, the human graders involved in this study did not receive standardised training before assessing the severity of diabetic retinopathy. A validated automatic deep learning software is a potentially reliable and sensitive tool in detecting diabetic retinopathy [11] and can overcome the issue of high variability among human graders. Our results showed that the automated algorithm of a validated deep learning software performed at a good sensitivity level for both non-urgent referable diabetic retinopathy (90.1%) and urgent referable diabetic retinopathy (85.7%). 

The accurate and timely diagnosis of diabetic retinopathy is of paramount importance in preventing vision loss and blindness in patients with diabetes mellitus [26]. However, there are currently limited data regarding the variability in diabetic retinopathy severity grading among doctors, as well as regarding the comparison of human graders to automated deep learning diabetic retinopathy screening software. Misclassification among trained graders was reported to be as high as 22% in a diabetic retinopathy screening program [19]. Comparisons of diabetic retinopathy severity grading with fundus photography by ophthalmic health personnel and ophthalmologists have only reported fair to moderate agreement [25,27,28]. However, the utilisation of an automated deep learning diabetic retinopathy screening software holds promising potential in addressing the limitations associated with human graders [26]. By leveraging such software, both intra-grader and inter-grader variability can potentially be reduced, especially if the software is validated in accordance with population-specific characteristics [26]. Although the agreement between the automated deep learning diabetic retinopathy screening software with the reference (ground truth) was only classified as good, the receiver operating characteristic curves showed good results of an area under the curve (ranging from 0.714 to 0.879) with a good level of sensitivity for non-referable diabetic retinopathy, and a very good level of sensitivity for both the non-urgent referable diabetic retinopathy and urgent diabetic retinopathy referral groups. 

According to Gargeya et al., a fully data-driven artificial intelligence-based grading algorithm can be used to effectively screen fundus photographs and reliably identify diabetic retinopathy with a high level of reliability [29]. Similarly, Limwattanayingyong et al.’s longitudinal study demonstrated that a deep learning-based system had higher sensitivity and positive predictive values than human graders did in both initial and follow-up diabetic retinopathy screenings [26]. Multiple studies have consistently reported the favourable attributes of automated deep learning diabetic retinopathy screening software, including its high accuracy, time efficiency, and cost-effectiveness, in comparison to human graders [26,30,31,32]. However, despite the enhancement of precision in diabetic retinopathy screening, several real-world challenges remain, including practical obstacles such as the use of mydriasis to enhance image quality, which has the potential to cause an angle closure attack [17]. Additionally, integrating this technology into healthcare systems raises concerns about the accuracy of diagnosis, and the governance of artificial intelligence in healthcare must adhere to established guidelines for fairness, transparency, trustworthiness, and accountability to protect the interests of all stakeholders [17].

A telemedicine diabetic retinopathy screening program using trained non-physician graders exists in Singapore, with results showing better time efficiency and cost-effectiveness than the existing model of family physician-based assessment of fundus photographs [19]. However, considering the projected increase in diabetes mellitus and the demand for more efficient screening methods, automated deep learning diabetic retinopathy screening software may be a more cost-effective tool for the future. The automated deep learning diabetic retinopathy screening software used in this study demonstrated high sensitivity for both the non-urgent referable diabetic retinopathy group (90.1%) and urgent referable group (85.7%). Although the agreement between the automated deep learning diabetic retinopathy screening software and the reference was only classified as good, the receiver operating characteristic analysis yielded positive outcomes, displaying good levels of sensitivity for non-referable diabetic retinopathy and remarkably high levels of sensitivity for both non-urgent referable diabetic retinopathy and urgent diabetic retinopathy referral groups. 

There are numerous commercially available diabetic retinopathy screening algorithms worldwide, including IDx-DR, RetmarkerDR, EyeArt, Singapore SERI-NUS, Google Inc, Bosch DR algorithm, Retinalyze, and Messidor-2 [12,17]. These screening algorithms, including the DiaRetDB algorithm utilised in this study, achieve a desirable equilibrium between high sensitivity and specificity. A low sensitivity in a test means that cases of diabetic retinopathy may be missed, and a lower specificity yields a significant number of false positives resulting in unnecessary referral to a tertiary centre. These screening algorithms have varying sensitivity and specificity in detecting diabetic retinopathy, ranging from 73.0% to 96.8% and 71.6% to 98.0%, respectively [12]. A direct comparison of available algorithms has been challenging due to variations in reference standards (either International Clinical Diabetic Retinopathy Disease Severity Scale or Early Treatment Diabetic Retinopathy Study Grading), grader experience and capabilities, the algorithm output of the system (diabetic retinopathy present/absent; referable diabetic retinopathy present/absent; no diabetic retinopathy/referable diabetic retinopathy/sight-threatening diabetic retinopathy; or others) and the definitions of referable or non-referable diabetic retinopathy [12,33].

This study provides valuable insights into the variability of diabetic retinopathy severity grading among human graders with different levels of experience and expertise, as well as between human graders and automated deep learning diabetic retinopathy screening software, using macula-centred fundus photographs in a real-world non-research setting. The study’s limitations include its small dataset consisting of only two hundred diabetic retinopathy macula-centred fundus photographs. However, it is important to acknowledge that this was a deliberate choice in order to enhance the accuracy of the qualitative analysis, preventing a premature extrapolation of findings, and identifying subtle differences that might not be clinically significant. Another limitation is the potential for the inaccurate assessment of clinically significant macular oedema using two-dimensional fundus photographs and the small sample size for proliferative diabetic retinopathy cases. Several studies assessing the application of artificial intelligence software for diabetic retinopathy have utilised the surrogate marker of clinically significant macular oedema, which is typically defined as the presence of hard exudates within a one-disc diameter from the centre of the macula [34]. However, while this marker is suitable for referral purposes, it may not provide an adequate determination of the overall severity of clinically significant macular oedema. The observed variability between human graders and automated deep learning diabetic retinopathy screening software highlights the importance of combining human expertise with technological advancements in diabetic retinopathy screening and diagnosis. Another important finding of this study is the potential value of incorporating macular optical coherence tomography imaging for the analysis of clinically significant macular oedema concurrent with the grading of two-dimensional fundus photographs. Although using artificial intelligence software for diabetic retinopathy has shown promise, the limitations of using a surrogate marker for clinically significant macular oedema underscore the importance of further research into more comprehensive diagnostic approaches.

## 5. Conclusions

This study highlights the significant wide range of inter-observer and intra-group variability among medical professionals in diabetic retinopathy screening. An automated deep learning diabetic retinopathy screening software that is validated for a specific population has the potential to be a valuable tool for the mass screening of diabetic retinopathy. This technology could increase the efficiency and cost-effectiveness of current screening programs operated mainly by trained human graders or ophthalmologists.

## Figures and Tables

**Figure 1 healthcare-11-01697-f001:**
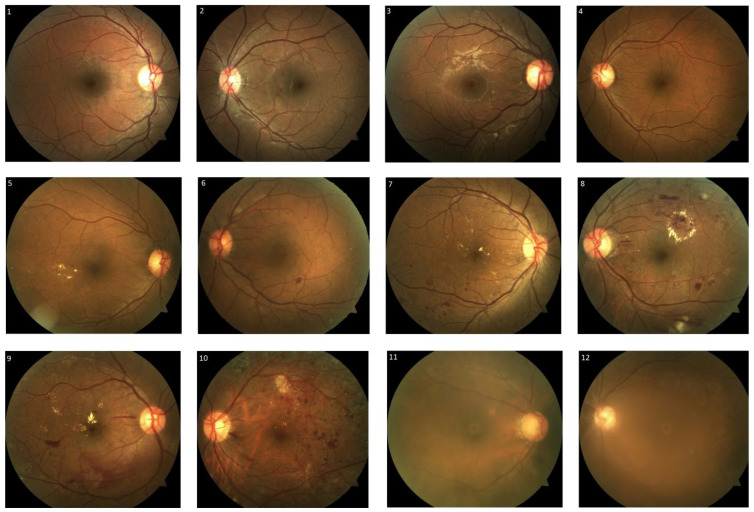
Examples of fundus photographs graded as no diabetic retinopathy (**1**,**2**), mild non-proliferative diabetic retinopathy (**3**,**4**), moderate non-proliferative diabetic retinopathy (**5**,**6**), severe non-proliferative diabetic retinopathy (**7**,**8**), proliferative diabetic retinopathy (**9**,**10**) and poor image quality (**11**,**12**).

**Figure 2 healthcare-11-01697-f002:**
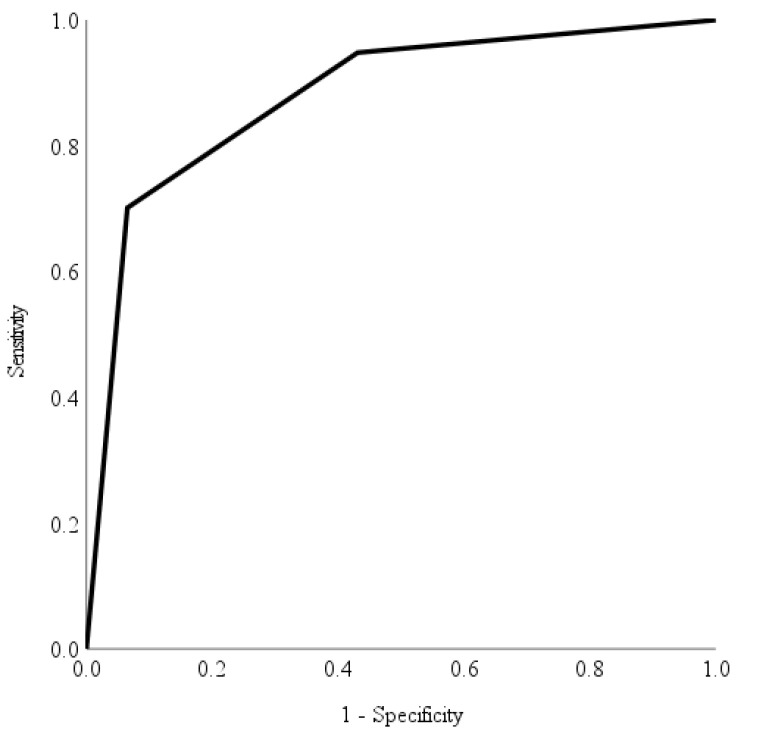
The receiver operating characteristic curve analysis of automated deep learning diabetic retinopathy screening software with reference for non-referable diabetic retinopathy. The area under curve was 0.879 (95% confidence interval: 0.829 to 0.929) with a sensitivity of 70.1%, specificity of 93.5%, positive predictive value of 87.1% and negative predictive value of 83.3%.

**Figure 3 healthcare-11-01697-f003:**
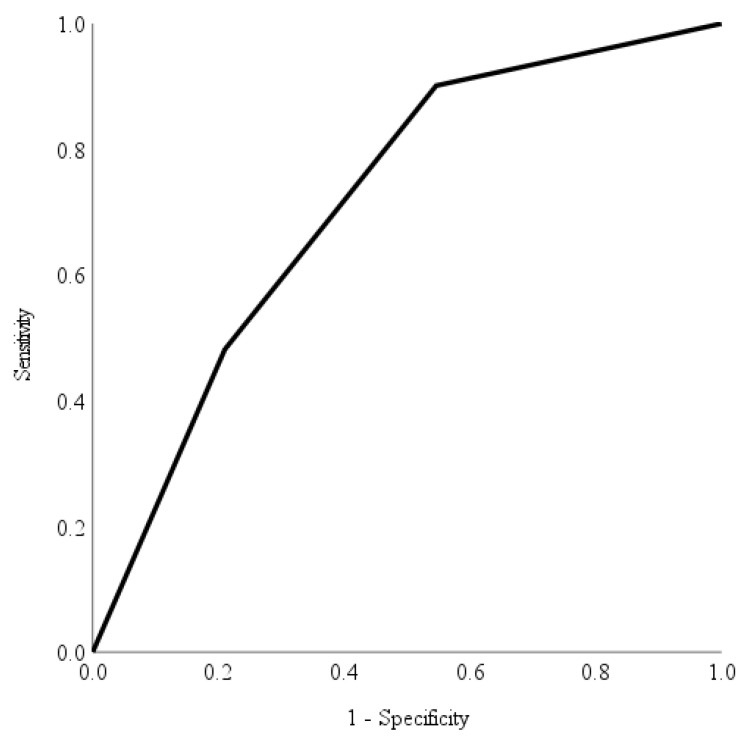
The receiver operating characteristic curve analysis of automated deep learning diabetic retinopathy screening software with reference for non-urgent referable diabetic retinopathy. The area under curve was 0.714 (95% confidence interval: 0.643 to 0.785) with a sensitivity of 90.1%, specificity of 45.4%, positive predictive value of 54.9% and negative predictive value of 87.1%.

**Figure 4 healthcare-11-01697-f004:**
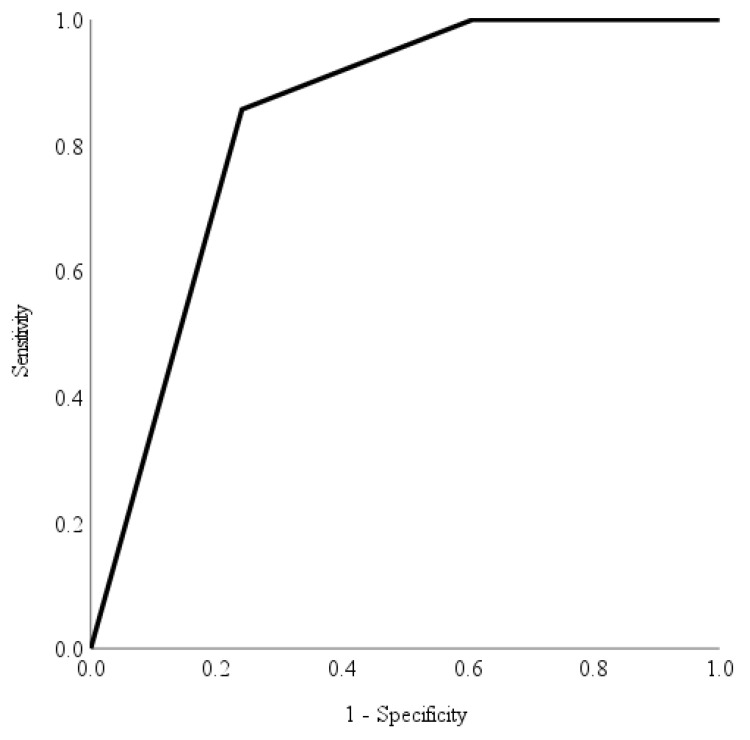
The receiver operating characteristic curve analysis of automated deep learning diabetic retinopathy screening software with reference for urgent referable diabetic retinopathy. The area under curve was 0.836 (95% confidence interval: 0.779 to 0.893) with a sensitivity of 85.7%, specificity of 75.9%, positive predictive value of 48.6% and negative predictive value of 95.2%.

**Table 1 healthcare-11-01697-t001:** Characteristics of graders involved in the current study.

Qualification of Graders	Year(s) of Experience in Grading Diabetic Retinopathy Photographs	Number of Grader(s)
Ophthalmology Group		
Retinal specialist	1 with 15 years’ experience; 2 with 10 years’ experience	3
Ophthalmology resident of board-certified program	4	3
Non-ophthalmology Group		
Family medicine physician	1 with 5 years’ experience; 2 with 8 years’ experience	3
Fifth-year undergraduate medical student	0	2

**Table 2 healthcare-11-01697-t002:** Inter-observer and intra-group agreements, presented in Gwet’s agreement coefficient values (S1, S2 and S2: retinal specialists; R1, R2 and R3: ophthalmology residents; M1 and M2: undergraduate medical students; F1, F2 and F3: family physicians; SW: automated deep learning diabetic retinopathy screening software).

Graders	Inter-Observer Gwet’s Agreement		Intra-Group Gwet’s Agreement	
	Coefficient Values	Level of Agreement	Coefficient Values	Level of Agreement
Ophthalmology Group				
S1	0.848	Very Good Agreement	0.628	Good Agreement
S2	0.720	Good Agreement		
S3	0.841	Very Good Agreement		
R1	0.677	Good Agreement	0.655	Good Agreement
R2	0.574	Moderate Agreement		
R3	0.549	Moderate Agreement		
Non- Ophthalmology Group				
F1	0.608	Good Agreement	0.536	Moderate Agreement
F2	0.377	Fair Agreement		
F3	0.613	Good Agreement		
M1	0.489	Moderate Agreement	0.446	Moderate Agreement
M2	0.205	Fair Agreement		
SW	0.628	Good Agreement		

## Data Availability

Public access to data is restricted due to ethical considerations imposed by the National Health Group Domain Specific Review Board (NHG-DSRB).

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
