# Peer review of "Variability in Grading Diabetic Retinopathy Using Retinal Photography and Its Comparison with an Automated Deep Learning Diabetic Retinopathy Screening Software"

_healthcare, 2023, doi:10.3390/healthcare11121697_

Round 1

Reviewer 1 Report

This paper proposed automated deep-learning software to classify diabetic retinopathy.

However, there are several comments from reviewers.

1. In line 16, "deep learning software (ADLS). Methods: 200 macula centered fundus photographs were graded", it is indicated typo. Please fix it.

2. Please consistency the font size on line 33.

3. Please refer to the instruction for authors "In the text, reference numbers should be placed in square brackets [ ], and placed before the punctuation; for example [1], [1–3] or [1,3]. For embedded citations in the text with pagination, use both parentheses and brackets to indicate the reference number and page numbers; for example, [5] (p. 10). or [6] (pp. 101–105).". The current version of this manuscript is not following the MDPI template.

4. In line 120, "Our own retinal" is an incorrect English structure. Please use the English editing service offered by the MDPI journal to improve English quality before submission.

5. What is the meaning of the term automated deep learning software? Are the authors creating software to run deep learning to use easily? Could the author show the interface of the software?

6. The introduction and method are confusing. Please explain more clearly.

Author Response

  1. In line 16, "deep learning software (ADLS). Methods: 200 macula centered fundus photographs were graded", it is indicated typo. Please fix it.

Response 1: Thank you for bringing the typo to our attention. We have made the necessary corrections as shown below:

Page 1, line18: “two hundred macula-centred fundus photographs were graded”

  1. Please consistency the font size on line 33.

Response 2: We have ensured consistency in the font size throughout the manuscript

  1. Please refer to the instruction for authors "In the text, reference numbers should be placed in square brackets [ ], and placed before the punctuation; for example [1], [1–3] or [1,3]. For embedded citations in the text with pagination, use both parentheses and brackets to indicate the reference number and page numbers; for example, [5] (p. 10). or [6] (pp. 101–105).". The current version of this manuscript is not following the MDPI template.

Response 3: We have amended the manuscript to follow the MDPI template.

  1. In line 120, "Our own retinal" is an incorrect English structure. Please use the English editing service offered by the MDPI journal to improve English quality before submission.

Response 4: We would like to express our apologies for the previous incorrect English structure in our statement. We have now made the necessary correction of the statement from “Our own retinal dataset and publicly available database DiaRetDB were used for the DR lesion detection and classification training” to “The training and validation were done using a dataset consisting of 30,000 images and DiaRetDB database”.

Additionally, we have improved the quality of the manuscript.

  1. What is the meaning of the term automated deep learning software? Are the authors creating software to run deep learning to use easily? Could the author show the interface of the software?

Response 5: In our study, deep learning software refers to a type of artificial intelligence program that utilises deep neural networks to analyse images. It mimics the way ophthalmologists determine the severity of diabetic retinopathy on colour retina photographs. The software enables automation via a user interface that allows photographs to be uploaded; processed with the deep learning software; and presents the results/recommendations to the user on the same platform. The software was created by the Commonwealth Scientific and Industrial Research Organisation and has been validated outside Singapore. To access the information and data of the software, one can search diaretdb database using the following link: https://www.kaggle.com/datasets/nguyenhung1903/diaretdb1-v21

The following figures show the interface of the software:

  1. The introduction and method are confusing. Please explain more clearly.

Response 6: We appreciate your feedback on the clarity of the introduction and method. We have edited and reformatted the sections to make them clearer.

Reviewer 2 Report

1. Title needs revision.

2. Novelty has to be highlighted in teh abstract

3. Recent works related to the study can  be introduced after the intro section along with implications.

4. How to validate the model?

5. Any comparison with recent works will be beneficial.

6. Rigorous experimenation is required

7. Strong conclusion recommended

8. Proof reading recommended

Author Response

  1. Title needs revision.
  • Respond 1: Thank you for your suggestion to revise the title. We have made the necessary changes and edited the original title "Agreement in grading of diabetic retinopathy severity using retinal photography and its comparison with automated deep learning software" to "Variability in grading severity of diabetic retinopathy through retinal photography and its comparison against validated automated deep learning software".

  1. Novelty has to be highlighted in teh abstract
  • Respond 2: We appreciate your feedback regarding the abstract. We have endeavored to emphasize the novelty of our study in this section.

  1. Recent works related to the study can be introduced after the intro section along with implications.
  • Respond 3: As this is our first collaborative study between Australia and Singapore, we have not yet progressed to the next stage of our research. Therefore, we regret that we are unable to incorporate recent works related to the study in the introduction at this time.

  1. How to validate the model?
  • Respond 4: Thank you for your query regarding how we validated our software. The software was developed by the Commonwealth Scientific and Industrial Research Organisation in Australia and has been validated. To obtain information and data on the software, you can refer to the diaretdb database.

  1. Any comparison with recent works will be beneficial.
  • Respond 5: We appreciate your suggestion to compare our work with recent studies. Unfortunately, we are unable to provide additional new data related to this comparison at this time.

  1. Rigorous experimenation is required
  • Respond 6: We acknowledge your critique of our experimentation and have taken measures to ensure greater rigor in the study.

  1. Strong conclusion recommended
  • Respond 7: Thank you for highlighting the need for a stronger conclusion. We have reviewed and revised our conclusions to make them more robust.

  1. Proof reading recommended
  • Respond 8: We are grateful for your recommendation to proofread our work. We have completed proofreading in this submission.

Round 2

Reviewer 1 Report

Thank you for the response,

Based on #5 comment, the figure shown on the comment should be included on the manuscript since it is comment from reviewer for improve paper quality. It is useless if the comment only for answer the reviewer without writing on the manuscript.

Reviewer 2 Report

No more suggestions

Author Response

There is no comment to address.